# Sensitivity of aerosol optical properties to the aerosol size distribution over central Europe and the Mediterranean Basin

Laura Palacios-Peña[1], Jerome D. Fast[2], Enrique Pravia-Sarabia[1], and Pedro Jiménez-Guerrero[1,3]

[1]Physics of the Earth, Regional Campus of International Excellence (CEIT) "Campus Mare Nostrum", University of Murcia, Spain.
[2]Pacific Northwest National Laboratory, Richland, WA, USA
[3]Biomedical Research Institute of Murcia (IMIB-Arrixaca), Spain.

**Correspondence:** Pedro Jiménez-Guerreo (pedro.jimenezguerrero@um.es)

**Abstract.** The size distribution of atmospheric aerosols plays a key role for understanding and quantifying the uncertainties related to aerosol-radiation and aerosol-cloud interactions. These interactions ultimately depend on the size distribution through optical properties (as aerosol optical depth, AOD) or cloud microphysical properties. Hence, the main objective of this contribution is to disentangle the impact of the representation of aerosol size distribution on aerosol optical properties over central Europe and particularly over the Mediterranean Basin during a summertime aerosol episode. To fulfill this objective, a sensitivity test has been conducted using the coupled chemistry-meteorology WRF-Chem model. The test modified the parameters defining a log-normal size distribution (geometric diameter and standard deviation) by 10, 20 and 50 %. Results reveal that the reduction in the standard deviation of the accumulation mode leads to the largest impacts in AOD due to a transfer of particles from the accumulation mode to the coarse mode. A reduction in the geometric diameter of the accumulation mode has also an influence on AOD representation since particles in this mode are assumed to be smaller. In addition, an increase in the geometric diameter of the coarse mode produces a redistribution through the total size distribution by relocating particles from the finer modes to the coarse.

## 1 Introduction

Aerosol size distribution is, among others, a key property of atmospheric aerosols which largely determines how they interact with radiation and clouds. Aerosol optical properties, as the scattering phase function, single scattering albedo, or aerosol optical depth (AOD), strongly depend on the aerosol size distribution (Eck et al., 1999; Haywood and Boucher, 2000; Romakkaniemi et al., 2012; Obiso et al., 2017; Obiso and Jorba, 2018). In this sense, AOD importantly influences aerosol-radiation interactions (ARI) and their associated radiative forcing (Boucher and Anderson, 1995; Boucher et al., 1998; Myhre and Stordal, 2001).

On the other hand, atmospheric aerosols influence climate forcing through aerosol-clouds interactions (ACI). These interactions produce an impact on clouds and precipitation which is connected with the number concentration of particles, which can act as cloud condensation nuclei (CCN) and ice nuclei (IN). Ultimately, these condensation nuclei depend on the aerosol size distribution and composition (Andreae and Rosenfeld, 2008; Romakkaniemi et al., 2012).

25    In this sense, the representation of aerosol processes in meteorological or climate models presents a high uncertainty (Boucher et al., 2013). Particularly, modelling aerosol size distribution introduces a noticeable uncertainty in chemistry transport models (Tegen and Lacis, 1996; Claquin et al., 1998). Three different approaches, deeply described in the Methodology section, are usually employed for aerosol models: (1) the bulk approach (only the aerosol mass concentration is computed); (2) the modal approach (multiple superposed modes); and (3) the sectional representation (aerosol size distribution discretized 30 into classes or bins).

These three approaches for representing aerosols are included in the WRF-Chem model, which is the coupled chemistry-meteorology model used in this work. The sectional approach is used by the Model for Simulating Aerosol Interactions and Chemistry (MOSAIC; Zaveri and Peters, 1999) and a simple scheme for volcanic ash (Stuefer et al., 2013). With respect to the modal approach, the schemes available within WRF-Chem are the Modal Aerosol Dynamics Model for Europe (MADE; 35 Ackermann et al., 1998) and the Modal Aerosol Model from CAM5 (MAM; Liu et al., 2012). Finally, the Goddard Global Ozone Chemistry Aerosol Radiation and Transport (GOCART; Ginoux et al., 2001; Chin et al., 2002) uses the bulk approach.

Some of these schemes have been widely applied for the study of aerosol optical properties and their uncertainty. In this sense, the evaluation of aerosol optical properties as represented by the MOSAIC approach has been conducted by Barnard et al. (2010) or Lennartson et al. (2018) to analyze the diurnal variation of AOD. Ghan et al. (2001) went a step beyond and 40 evaluated the radiative impact of including coupled aerosol-cloud-radiation processes. In addition, some contributions had the objective of assessing the representation of aerosol optical properties and their uncertainties using MOSAIC together with other schemes, mainly MADE (Zhao et al., 2010, 2011, 2013; Balzarini et al., 2015; Yang et al., 2018; Saide et al., 2020). The GOCART scheme has also been used for this aim. For example, LeGrand et al. (2019) compared the Air Force Weather Agency (AFWA) dust emission scheme within GOCART to other dust emission schemes available in WRF-Chem and their 45 skills for representing AOD. In this contribution, the need for tuning the model in order to get a reasonable simulation of AOD for each location and/or event was pointed out based on the results of Bian et al. (2011); Dipu et al. (2013); Kumar et al. (2014); Jish Prakash et al. (2015); Zhang et al. (2015); Kalenderski and Stenchikov (2016); Hu et al. (2020); among others. All those works evaluated the representation of AOD depending on the approach followed by the aerosol scheme. However, this contribution evaluates the uncertainty associated to the representation of the aerosol size distribution when estimating aerosol 50 optical properties.

In addition to the complexity of characterizing adequately the representation of aerosols, the complexity of the target area where the model is applied (e.g. orography, emissions or chemical transport) hampers the correct representation of atmospheric aerosols. Particularly, Europe (and specifically the Mediterranean Basin) is one of the most sensitive regions to aerosol forcing (Giorgi, 2006). Not only anthropogenic aerosols are presented over the Mediterranean Basin, but also sea salt, desert dust and 55 biomass burning (mainly from summer wildfires). Particularly in summer, when the aerosol forcing is larger (e.g. Charlson

et al., 1992), the role of ARI and ACI over this area is crucial (Papadimas et al., 2012) and much more important than over central Europe (Andreae et al., 2002). This is because of the complex terrain and the geographical location of the Mediterranean area, in addition to the processes undergone by aerosol particles (e.g. intense formation, accumulation and recirculation; Millán et al., 1997; Pérez et al., 2004; Querol et al., 2009).

Due to the important effects of aerosol size distribution and because previous works have highlighted the misrepresentation of size distribution by models (e.g. Palacios-Peña et al. (2017), Palacios-Peña et al. (2018) or Palacios-Peña et al. (2019a)), this contribution analyzes the sensitivity of aerosol optical properties to the representation of aerosol size distribution over central Europe, and particularly over the Mediterranean Basin. For that purpose, a modelling approach has been used for a typical case study during summertime, in order to estimate the sensitivity of AOD to the log-normal distribution parameters (geometric

diameter and standard deviation) that characterize the aerosol size distribution in the simulations conducted. These parameters will finally influence the representation of ARI and ACI in coupled chemistry-meteorology models. Section 2 describes the methodological aspects of this contribution and the model setup. Section 3 depicts the results found; and Section 4 discusses and summarizes the results.

## 2    Methodology

The methodology relies on a sensitivity test, carried out using the on-line WRF-Chem model, whose objective is the analysis of the response of AOD to modifications in the aerosol size distribution. The test modified the geometric diameter (DG) and the standard deviation (SG) of the log-normal function representing the aerosol size distribution. Each parameter has been modified by $\pm 10$, 20 and 50 % with respect to its initial value in each of the three modes represented (Aitken, accumulation and coarse).

In order to elucidate how important the changes of AOD are in each experiment and to avoid the analysis of random changes, a Kolmogorov-Smirnov test (Stephens, 1974) has been applied. This is a non-parametric test which can be used to compare two samples by their probability distribution (Stephens, 1974). In this case, all the experiments are compared with the base case to rank the importance of the changes produced. The most important cases will be selected as reference cases for a further discussion leading to disentangle the causes of the changes in AOD provoked by the modifications made to the size distribution

parameters.

### 2.1    Case study: 4-9 July 2015

The case study selected here covers an extended episode between 4 and 9 of July, 2015. The synoptic overview of the meteorological conditions has been widely presented in Palacios-Peña et al. (2019b). Nonetheless, a brief summary of the meteorological episode selected is presented here.

During this episode, the development of an omega-blocking situation takes place, with low pressures located over western England. The episode presents a high stability over the Mediterranean Basin with a high aerosol load, fire emissions in the target area, and a strong dust outbreak induced by the penetration of warm air and dust from northwestern Africa towards the

western Mediterranean Basin and northern Europe (Nabat et al., 2015). The weakening of this synoptic situation results in a cyclonic circulation of the air over the western Mediterranean (see Figure 1 in Supplementary Material, Palacios-Peña et al., 2019b). The choice of this episode reveals the crucial role of aerosols from different sources over the Mediterranean Basin, whose forcing is even stronger during summertime.

## 2.2 Model setup

The Weather Research Forecast model coupled with Chemistry (Grell et al., 2005, WRF-Chem;) version 3.9.1.1 was used in this work. This fully coupled on-line model represents ARI and ACI by allowing the simultaneous treatment of meteorology and gases and aerosols emissions, transport, mixing, and chemical transformation.

As aforementioned, the case study selected here trusts on an extended episode evaluated in Palacios-Peña et al. (2019b). The model setup for all the experiments is the same as for that contribution. However, a brief summary of the configuration and parameterizations is included in Table 1.

The target domain covers central Europe and the Mediterranean Basin with a resolution of $\sim 0.15^\circ$ ($\sim 16.7$ km, Figure 2), but this target domain is the inner domain (D3) of a nested run that allows to capture the total desert dust contribution from the Sahara Desert (Fig. 1). For that purpose, a parent domain covering the main areas of desert dust emissions (located around 15 °N) was used. The other domains were built by one-way nesting with a nesting ratio of 1:3 with respect to the larger domain. Thus, the parent domain has a spatial resolution of $1.32^\circ$ (150 km) and the second of $0.44^\circ \sim 50$ km. Vertical resolution presents 48 uneven layers with the highest resolution at the bottom. The top of the atmosphere at set at 50 hPa.

Meteorological initial and boundary conditions for the outer domain were provided by the ERA-Interim reanalysis (Dee et al., 2011). The WRF-Chem option for idealized gases and aerosol profile has been chosen as chemical boundary conditions. Anthropogenic emissions were provided by the Emissions Database for Global Research-Task Force on Hemispheric Transport of Air Pollution (EDGAR-HTAP) project (http://edgar.jrc.ec.europa.eu/htap.php; Janssens-Maenhout et al., 2012). Biomass burning emission data have been estimated from the Integrated monitoring and modelling system for wild-land fires (IS4FIRES; Sofiev et al., 2009; Soares et al., 2015). Both have been adapted to chemical species in WRF-Chem following Andreae and Merlet (2001) and Wiedinmyer et al. (2011); and plume rise calculation was on-line estimated by WRF-Chem. Biogenic emissions are on-line coupled with WRF-Chem by using the Model of Emissions of Gases and Aerosol from Nature (MEGAN; Guenther et al., 2006). Finally, dust (Ginoux et al., 2001) and sea salt GOCART (Chin et al., 2002) emissions were on-line estimated by WRF-Chem.

The GOCART aerosol scheme in WRF-Chem does not allow a full coupling of aerosol-cloud interactions. For instance, convective wet scavenging (conv_tr_wetscav), in-cloud wet scavenging and cloud chemistry are not available. However, in those simulations denoted as ACI, the Morrison microphysics (used in this contribution) acts as a double-moment scheme meanwhile in the rest of the simulations it works as a single-moment microphysics scheme. This latter approach is unsuitable for assessing aerosol-cloud interactions because it does not include a prognostic treatment of droplet number. Hence, the ACI configuration allows a double-moment microphysics with greater flexibility when representing size distributions and hence microphysical process rates (Palacios-Peña et al., 2020). When the double-moment scheme is activated, a prognostic droplet

**Table 1.** Model setup for the experiments.

| Mechanism | Option | Reference |
|---|---|---|
| *Physic configuration* | | |
| Microphysics | Morrison | Morrison et al. (2009) |
| Short wave radiation | Rapid Radiative Transfer Model | Iacono et al. (2008) |
| Long wave radiation | (RRTM) | |
| Planetary boundary layer | Yonsei University scheme | Hong et al. (2006) |
| Cumulus | Grell-Freitas ensemble | Grell and Freitas (2014) |
| Soil | Noah | Tewari et al. (2004) |
| *Chemical configuration* | | |
| Gas-phase | Regional Atmospheric Chemistry Mechanism | Stockwell et al. (1997) |
| | Kinetic Pre-Processor (RACM-KPP) | Geiger et al. (2003) |
| Aerosol | Global Ozone Chemistry Aerosol Radiation and | Ginoux et al. (2001) |
| | Transport model (GOCART) | Chin et al. (2002) |
| Dust | GOCART emissions | Ginoux et al. (2001) |
| Sea Salt | GOCART emissions | Chin et al. (2002) |
| Photolysis | Fast-J | Wild et al. (2000) |
| Dry deposition | | Wesely (1989) |
| Wet deposition | Grid-scale calculated | |
| Aerosol-radiation interactions | On | |
| Aerosol-cloud interactions | On | |

number concentration using gamma functions and mixing ratios of cloud ice, rain, snow, graupel/hail, cloud droplets and water vapour are estimated (Morrison et al., 2009). Finally, the interaction of cloud and solar radiation with the Morrison microphysics scheme is implemented in WRF-Chem (Skamarock et al., 2008). Therefore, droplet number will affect both the
droplet mean radius and the cloud optical depth calculated by the model.

     The aerosol size distribution represents the number (N), mass (M), or volume (V) of particles as a function of diameter (d; Seinfeld and Pandis, 2006). Commonly, the aerosol size distribution is represented as a function of the logarithm of the diameter. Thus, the total number, mass, or volume of aerosol particles is the integral of the diameter over the size distribution function (Buseck and Schwartz, 2003).
When the aerosol size distribution is modeled, three different approaches are commonly used (Boucher, 2015): (1) First, a bulk approach in which only the aerosol mass concentration is computed. A constant size distribution is assumed and there is not a representation of a mixing state. That leads to a simple and computationally cheap approach. (2) A more complex approach

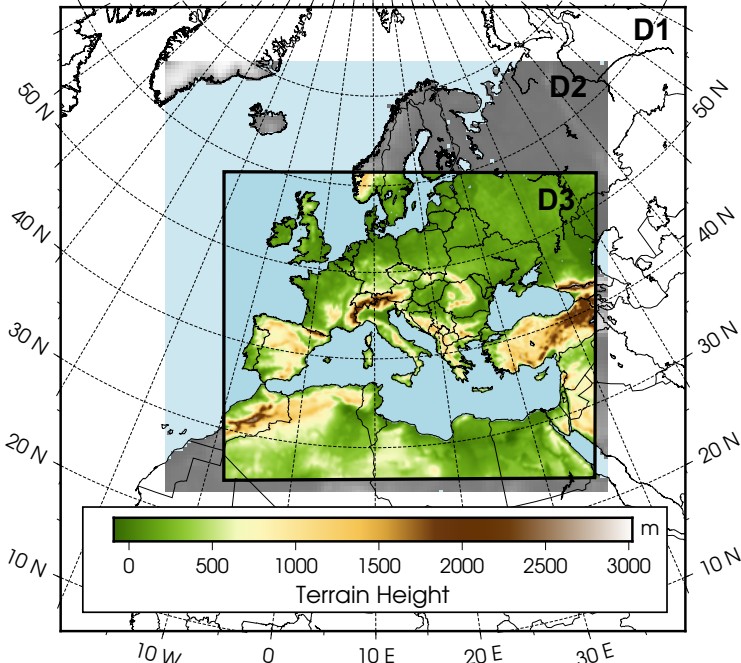

**Figure 1.** Target (D3) and nested domains (D1 and D2). Adapted from Palacios-Peña et al. (2019b).

uses multiple superposed modes, which are typically represented by a log-normal distribution described by a fixed mean and variance. The accuracy of this approach lies in the correct choice of these two parameters. (3) Last, the most computationally
expensive approach is the sectional representation, which consists in discretizing the aerosol size distribution into $n$ classes or bins of diameter ,where the concentration of aerosols in each bin follows the conservation equation and then the number, mass, or volume in each bin is predicted.

A log-normal approach is typically employed in chemistry transport models for aerosol size distribution, since this approach fits the observed aerosol size distribution reasonably well and its mathematical form is convenient for dealing with the moment
distribution. In this approach, all of the moment distributions are log-normal and present the same geometric mean diameter and geometric standard deviation, parameters which determine the log-normal distribution (Hinds, 2012). One of the most common log-normal size distributions used is that described in Heintzenberg (1994, Eq. 1), which has also been employed in this contribution. Equation 1 represents the zeroth moment of the particle size distribution, where $d_p$ is the particle diameter; $\sigma_g$ is the standard deviation of the distribution; $F_{m0}$ is the number concentration as diameter $d_{gN}$ (number median diameter).

$$\frac{dn}{d\ln d_p} = \frac{F_{m0}}{\sqrt{2\pi}\ln\sigma_g} \times exp\left[-\frac{(lnd_p - lnd_{g0})^2}{2(ln\sigma_g)^2}\right]$$
(1)

GOCART (Ginoux et al., 2001; Chin et al., 2002) includes a bulk approach for black carbon (BC), organic carbon (OC) and sulfate; and a sectional scheme for mineral dust and sea salt using Kok (2011) brittle fragmentation theory, a simple and cheap

computational approach. The selection of this scheme is conditioned by the fact that WRF-Chem version 3.9.1.1 only allows the simulation of desert dust and sea salt with this GOCART scheme.

The module of aerosol optical properties in WRF-Chem calculates optical properties from species estimated by the GOCART scheme. These properties depend on size and number distribution, composition and aerosol water. For a bulk approach as GOCART, bulk mass and number is converted into an assumed log-normal modal distribution, then dividing the mass into sections or bins ("$i$"). The parameters which define this log-normal distribution are the modified variables for the sensitivity test. Then the aerosol optical calculation follows the process described in Barnard et al. (2010). For each bin and each chemical

species ("$j$"), mass is converted to volume. Summing over all the species volume and assuming spherical particles, a diameter ($D$) is assigned to each bin. Therefore, the aerosol size distribution is defined by the number and the associated diameter for each bin. Aerosol water content depends on the relative humidity (RH) and the hygroscopicity factor of each species in the aerosol composition. Refractive indices are averaged, by Maxwell-Garnett approximation (Bohren and Huffman, 2007), among the compositions for each section in which mass has been divided. All particles within a size range are assumed to have the

same composition, although their relative fraction can differ among size ranges. Finally, an approximate version of the Mie solution (Ackerman and Toon, 1981) is used to estimate the absorption efficiency ($Q_{a,i}$), the scattering efficiency ($Q_{s,i}$) and the asymmetry parameter ($g_i$). Optical properties are computed by summing over the size distribution. Equation 2 shows an example for the estimation of the scattering coefficient ($\sigma_s$).

$$\sigma_s = \sum_{i=1}^{8bins} N_i Q_{s,i} \pi (\frac{D_i}{2})^2 \tag{2}$$

## 3   Results

First, the impacts of the sensitivity test on the representation of AOD are investigated. Then, the magnitude of these effects is analyzed by using the Kolmogorov-Smirnov test. Once the most relevant cases among all those run in the sensitivity test have been established, the causes of these changes are disentangled.

### 3.1   Effects on AOD representation

Figure 2 shows the AOD at 550 nm simulated by the base case (top row) and the differences between each of the experiments vs. the base case. This Figure only exhibits the results for the modification of 50 %, but the rest of the experiments are shown in the Supplementary Material, Figure 3). The reason to show only the 50 % experiment is the qualitatively similar spatial pattern of changes found for each experiment when modifying the parameters by 10, 20 and 50 %. Overall, the difference lies on the quantitatively larger changes on the latter experiment.

As aforementioned, the experiments consist in the modification of the geometric diameter (DG) and the standard deviation (SG) of the aerosol size distribution by 10, 20 and 50%. Taking this consideration into account, the experiments have been named indicating the sign of the modification ($L$ for a reduction/low and $H$ for an increase/high) and the percentage (10, 20

or 50), the variable of the size distribution modified (SG for the standard deviation and DG for the geometric diameter) and the mode in which the modification has been implemented ($ai$ for Aitken, $ac$ for accumulations and $co$ for coarse). Thus, the acronym for an experiment follows the this pattern: (L | H)(10 | 20 | 50)_(SG | DG)(ai | ac | co). For example, L50_SGai indicates a 50% reduction (L) of the standard deviation (SG) in the Aitken mode (ai).

The top row in Figure 2 displays the hourly mean of AOD for the base case, averaged for the entire target period (from 4 to 9 July). As established by Palacios-Peña et al. (2019b), high AOD values over the western part of the Mediterranean Basin and central Europe were caused by a strong desert dust outbreak from the Sahara Desert. The AOD at the eastern part of the Mediterranean has high values because the outbreak reached that area at the end of the period. Palacios-Peña et al. (2019b) evaluated this base case against observations coming from Moderate Resolution Imaging Spectroradiometer (MODIS; and instruments onboard satellite) and the Aerosol Robotic Network (AERONET). The evaluation results (Figure 2 in the Supplementary Material) evinced negligible errors of the model over large areas, but an underestimation of AOD over the north of Germany and the central and western Mediterranean (around -0.2 to -0.4). Albeit this underestimation, the spatio-temporal mean bias error is -0.02 (when evaluated against both observational datasets, MODIS and AERONET) and the mean absolute errors is 0.16 (when assessed against MODIS) and 0.12 (when evaluated against AERONET). Palacios-Peña et al. (2019b) also pointed out that this underestimation over the central and western Mediterranean Basin turns into a larger overestimation when a coarser resolution is used due to a worsening in the representation of the dynamical patterns and thus, the dust transport.

The results of the sensitivity tests indicate that AOD is not very sensitive to the modification of the standard deviation of the Aitken mode (L50_SGai and H50_SGai). Identical results are found for the variation of the geometric diameter of the Aitken mode (L50_DGai and H50_DGai). In these experiments there is not a clear pattern in the response of AOD to the perturbations of the test; that is, low positive and negative changes (most of them above 0.05) alternate spatially. Higher differences (around and above >0.1) are found over some small areas close to the boundaries. Temporal and spatial differences range between -0.03 and -0.01, indicating that there is not a clear impact of the modification of size distribution for the Aitken mode on AOD levels.

Sensitivity experiments including modifications of the accumulation mode lead to larger spatial changes and with a greater spatial extension when compared to sensitivity tests modifying the representation of the Aitken mode. The experiment where the standard deviation of the accumulation mode decreases (L50_SGac) is the experiment with the highest impact on AOD. Over most of the domain the change is limited, > 0.1. These changes are negative over central Europe and the Iberian Peninsula (where AOD in the base case is > 0.3) and are positive over the eastern Mediterranean Basin. No significant patterns of change are visualized when increasing the standard deviation of the accumulation mode (H50_SGac), with changes in AOD under $\pm 0.05$. With respect to the experiments modifying the geometric diameter, negative changes with a temporal and spatial mean of -0.04 are found when the DG decreases by 50 % (in general, the model is considering particles in the accumulation mode smaller than in the base case). On the other hand, when DG increases, positive changes are widely observed (larger particles than in the base case). Despite this overall signal, positive and negative signals alternate spatially in both simulations.

Finally, when considering changes in the size distribution of the coarse mode, H50_DGco (geometric diameter increases by 50%) is the experiment with the strongest response. Negative variations are found over large parts of the domain, with values

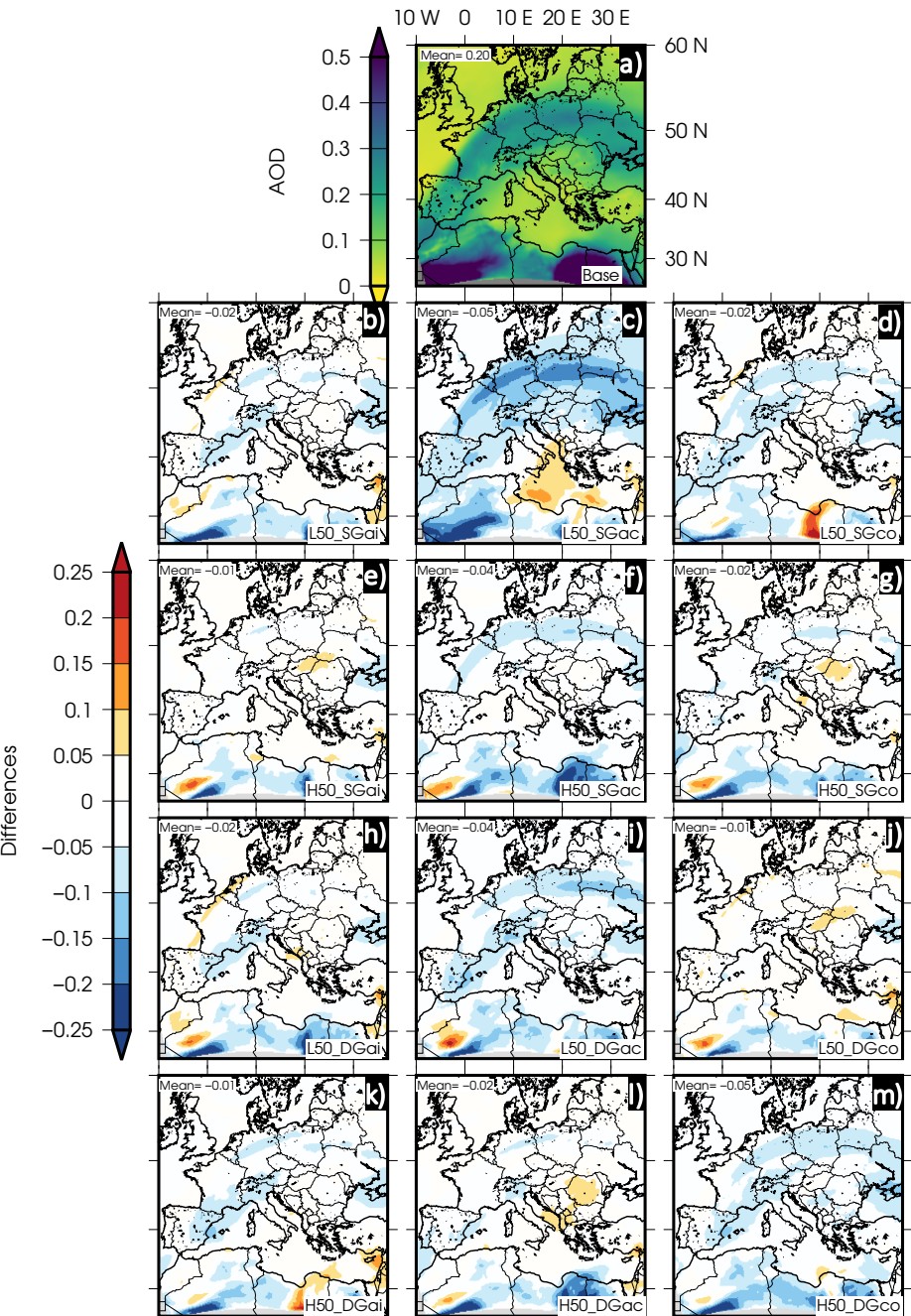

**Figure 2.** AOD at 550nm and differences for the sensitivity tests modifying the parameters by 50%. a) Base case; b) Aitken mode 50% reduction in SG; c) accumulation mode 50% reduction in SG; d) coarse mode 50% reduction in SG; e) Aitken mode 50% increase in SG; f) accumulation mode 50% increase in SG; g) coarse mode 50% increase in SG; h) Aitken mode 50% reduction in DG; i) accumulation mode 50% reduction in DG; j) coarse mode 50% reduction in DG; k) Aitken mode 50% increase in DG; l) accumulation mode 50% increase in DG; m) coarse mode 50% increase in DG.

up to -0.15 over smaller areas. However, when the DG decreases, the response oscillates between negative and positive values lower than 0.05. Analogously, the response to the modification of the standard deviation for the coarse mode does not show a clear pattern.

As mentioned previously, for all of the experiments, higher changes (above 0.1) are found close to the south boundary. This could be caused by the fact that the main natural sources of emissions are located over this area.

## 3.2 Significance of AOD changes

This section focuses on elucidating and ranking the importance of AOD changes for the sensitivity experiments, in order to select the experiments with the highest sensitivity. The physico-chemical causes behind those changes for the selected experiments will be disentangled later in Section 3.3.

For that purpose, Figure 3 displays the Probability Density Function (PDF) of the values of AOD at 550nm (for all cells and timesteps in the model) simulated by the base case (solid black line) and each of the experiments (dashed-red line). As in the previous section, this Figure only exhibits the results for the modification of 50 %, but the results for the rest of perturbations can be found in the Supplementary Material (Figure 4). The number in each panel represents the statistics obtained from the Kolmogorov-Smirnov test. The Kolmogorov-Smirnov is a non-parametric test used for the evaluation of the statistical similarity of the distribution between two datasets. The test is based in the assumed similarity of the Empirical Cumulative Distribution Function (ECDF) between two random samples. The maximum distance between both ECDFs, normally named as $D$, indicates how far both distributions are. In this contribution, the distribution of each experiment (dashed-red line) has been evaluated against the distribution of the base case (black line). The $p$-values represent the probability of values as extreme as those obtained for samples coming from the same distribution. Low $p$-values show a low probability of error when the null hypothesis is rejected and, thus, indicate that both samples do not provide from the same distribution (Sprent and Smeeton, 2016).

For all of the experiments the distance between the samples is statistically significant ($p-value$ close to 0) because of the high number of samples (cells) in each experiment. As all the spatio-temporal values are taken for statistical purposes, the number of samples is over 1,000,000. However, the distance varies for each experiment.

L50_SGac is the experiment with the highest $D$ (0.2277), meaning that this experiment presents the largest difference with respect to the base case. H50_DGco, with a distance of 0.1920, and L50_DGac, with 0.1648, are also experiments with a noticeable difference ($D$). H50_SGac, with a distance of 0.0891, also shows differences but not as important as for the former cases. The rest of the sensitivity cases present $D$ values lower than 0.05; that is, differences are small with respect to the base case, although statistically significant.

Similar results are found when the modification of 10 and 20 % are analyzed (Supplementary Material, Figure 4). These cases exhibit $D$ lower than for the modification of 50 % but higher than 0.05. The distance is lower as the magnitude of the modification decreases. For example, the L10_SGac distance is 0.0863, the L20_SGac is 0.1814 and the L50_SGac is 0.2270. This behaviour is repeated in the rest of the experiments, but with $D < 0.05$.

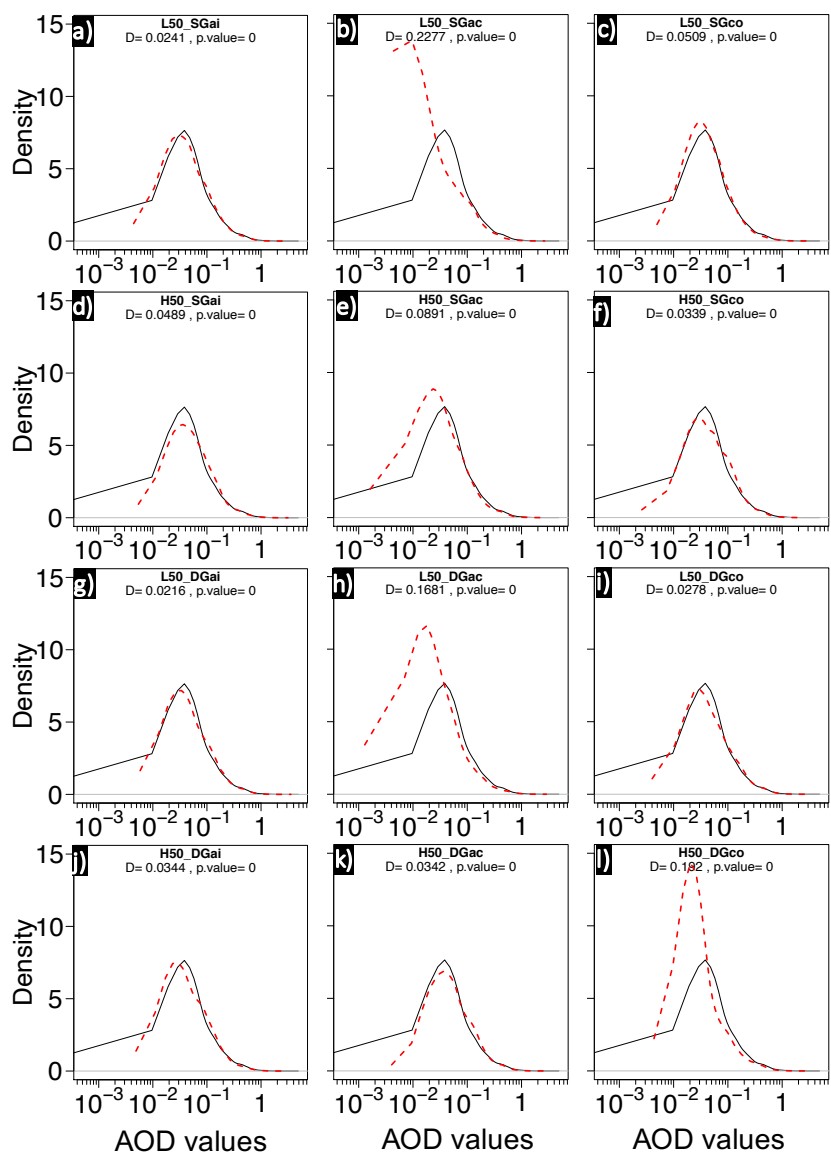

**Figure 3.** PDF of AOD values for the base case (black line) and each of the sensitivity test simulations at 50 % (dashed-red line). Values in Figures represent the results of the statistic from the Kolmogorov-Smirnov test. a) Aitken mode 50% reduction in SG; b) accumulation mode 50% reduction in SG; c) coarse mode 50% reduction in SG; d) Aitken mode 50% increase in SG; e) accumulation mode 50% increase in SG; f) coarse mode 50% increase in SG; g) Aitken mode 50% reduction in DG; h) accumulation mode 50% reduction in DG; i) coarse mode 50% reduction in DG; j) Aitken mode 50% increase in DG; k) accumulation mode 50% increase in DG; l) coarse mode 50% increase in DG.

These $D$ can be observed in the PDF shown in Figures 3. The second panel in the first row portraits the PDF for L50_SGac, showing a much higher peak (peak of density >10) than for the base case (peak of density <8). Moreover, the upper tail reaches AOD values <3 for the L50_SGac meanwhile the upper tail for the base case reaches AOD values <5. The response for the modification of 20 % and 10 % is similar. However, this latter shows larger AOD values in the upper tail and a peak of density lower (around 9).

A similar behaviour is exhibited by the H50_DGco and the L50_DGac experiments; but in the latter the upper tail reaches AOD >3. However, the PDF of this experiment does not respond analogously to other quantitative modifications (10 % and 20 %). H50_SGac is noticeable because its peak of density reaches values up to 9 and its upper tail up to 3; however, its distance is much lower (>0.1) than for the previously mentioned cases and decreases as the perturbation in the sensitivity experiment does.

In order to disentangle the causes for the results found in the sensitivity tests, the next section focuses in those cases where the distance in the Kolmogorov-Smirnov test with respect to the base case is >0.1. These are L50_SGac, L50_DGac and H50_DGco. Regarding other modifications, only the L20_SGac (Figure 4 in Supplementary Material) shows a distance higher than 0.1. The L10_SGac difference is not higher than 0.1 (because of the limited modification of 10 %) but the distance is the highest for this range of modifications, with a value of 0.09.

## 3.3 Disentangling the causes of AOD variations due to perturbations in the size distribution

Figure 4 displays the PM-ratio for the base case at 1000 hPa and the corresponding differences between the experiment at 50 % and the base case. This statistical figure is estimated as the ratio between $PM_{2.5}$ and $PM_{10}$ and is a proxy for the predominance of fine or coarse particles in the air mass. High values of PM-ratio imply a higher proportion of fine particles (usually with an anthropogenic origin) while low values of the ratio point to the presence of coarse particles (natural origin).

The PM-ratio of the base case (Figure 4 a) is close to zero in Africa because during this Saharan Desert dust episode coarse particles ($PM_{10}$) predominate over this area. When comparing the top row in Figure 4 and Figure 2, areas with high AOD levels match those areas with null PM-ratio due to the influence of desert dust. The high values of the ratio over the central Mediterranean Sea could be ascribed to the transport of dust particles. At the beginning of the episode, the dust outbreak reached the central Mediterranean (coarse particles from dust were modelled here), but as the episode developed, dust -and hence, $PM_{10}$ particles- moved eastwards and northwards, being the $PM_{2.5}$ concentrations higher over the Mediterranean Sea and the African coastline at the end of the target period.

For the selected cases, Figure 4 shows an inverse behavior of the PM-ratio with respect to AOD. The experiment decreasing the standard deviation in the representation of the accumulation mode (L50_SGac) presents a reduction up to -0.45 in the PM-ratio (that is, coarse particles become predominant) over the eastern Mediterranean Basin, which matches with the increase modeled for AOD. This behavior is also reproduced aloft (at 750 hPa, Figure 5 in the Supplementary Material) and is consistent with the sensitivities of 20 % and 10 % (Figure 6 in the Supplementary Material). However, for the other experiments the response is weaker.

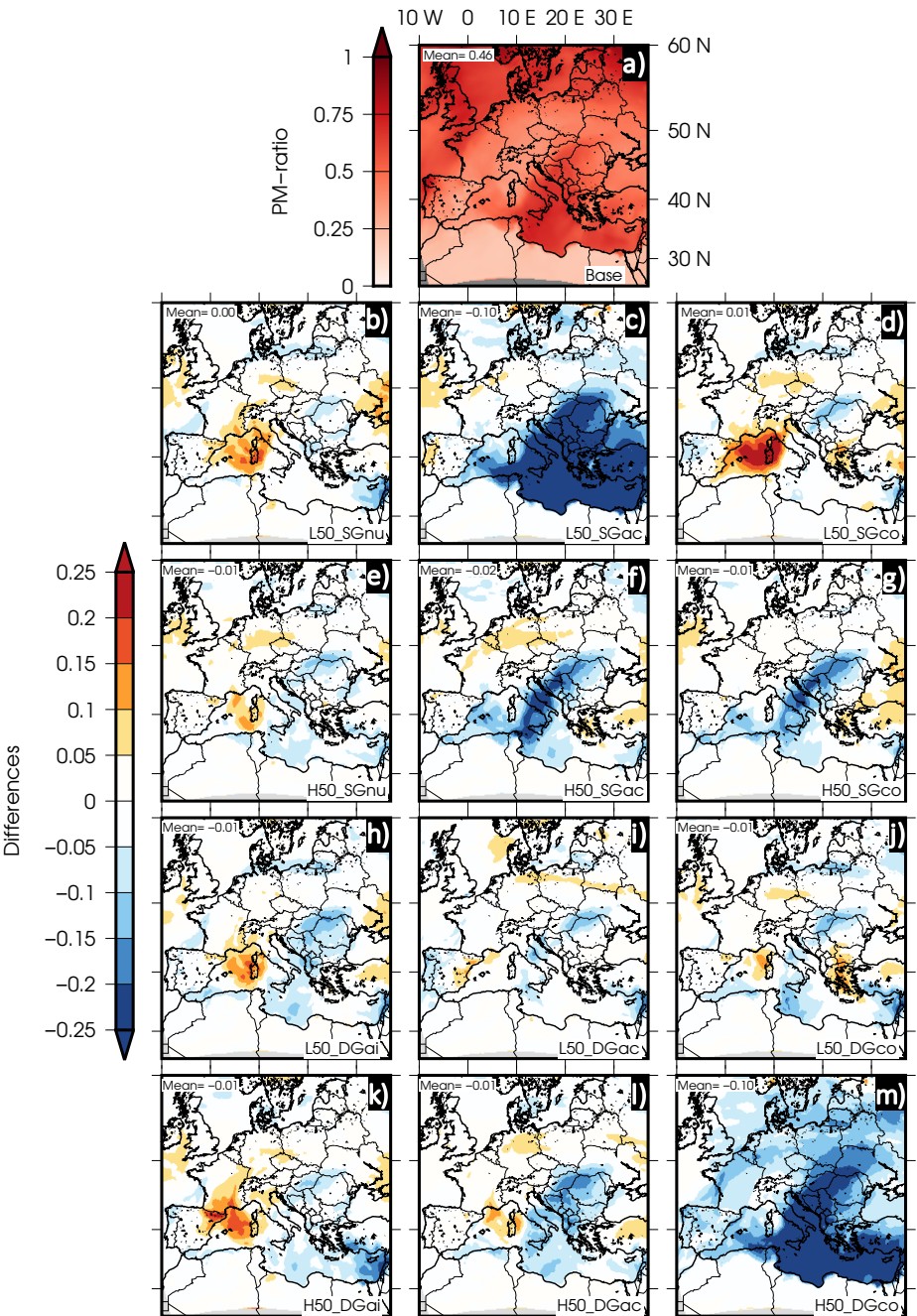

**Figure 4.** PM-ratio at 1000 hPa for the base case and differences for sensitivity simulations at 50 %. a) Base case; b) Aitken mode 50% reduction in SG; c) accumulation mode 50% reduction in SG; d) coarse mode 50% reduction in SG; e) Aitken mode 50% increase in SG; f) accumulation mode 50% increase in SG; g) coarse mode 50% increase in SG; h) Aitken mode 50% reduction in DG; i) accumulation mode 50% reduction in DG; j) coarse mode 50% reduction in DG; k) Aitken mode 50% increase in DG; l) accumulation mode 50% increase in DG; m) coarse mode 50% increase in DG.

The experiment decreasing the geometric diameter of the accumulation mode (L50_DGac) does not lead to large differences. This experiment shows a slight increase of the PM-ratio (around 0.05) over the western Mediterranean and central Europe, which points to a slight increase in fine particles; and a limited decrease (around -0.05) of the PM-ratio over Italy, Hungary and Romania.

Finally, the experiment increasing the geometric diameter of the coarse mode (H50_DGco) produces a different response. Albeit this experiment presents lower values (up to -0.25) for the PM-ratio over most of the target domain -hence highlighting the increase in the predominance of coarse particles-, AOD is also lower (in particular over central Europe).

In order to understand these changes, Figure 5 exhibits the total number concentration of particles at 1000 hPa in the Aitken+accumulation (summed) and coarse modes and the relative differences between the different experiments and the base case for the sensitivity tests modifying the parameters by 50 %. Figure 6 is similar to Figure 5 but for total mass concentration. Aloft particles (750 hPa, Figure 7 in the Supplementary Material), and sensitivities (20 % and 10 %) as well as non-relative differences are available for both total number and mass concentration in the Supplementary Material (Figures 8 to 17).

The experiment reducing the standard deviation in the accumulation mode (L50_SGacc) and its analogous lower modifications, L20_SGac and L10_SGac, show a similar response that becomes stronger the larger the modification is. Because of that, only L50_SGacc is analyzed in this contribution as it is representative of changes in SGacc. This experiment leads to a reduction in the total number concentrations (up to -80 % for the Aitken and accumulation modes and -60 % of the base case for the coarse particles) and total mass (up to -60 % of the base case for the Aitken and the accumulation modes and -40 % for the coarse mode) over the European continent for all the modes. However, a reduction in the total number concentration is found over the Mediterranean and over eastern and western areas for the Aitken and accumulations modes. An increase is depicted over the central and western Mediterranean for the coarse mode (higher than 80 % with respect to the base case). This increase in the total number concentration of the coarse mode could explain the decrease estimated for the PM-ratio and thus the increase of AOD as particles become larger. The reduction in both modes explains the observed decrease in AOD as the number and mass of particles decrease, which does not lead to modifications in PM-ratio because the total number concentration decreases in both modes.

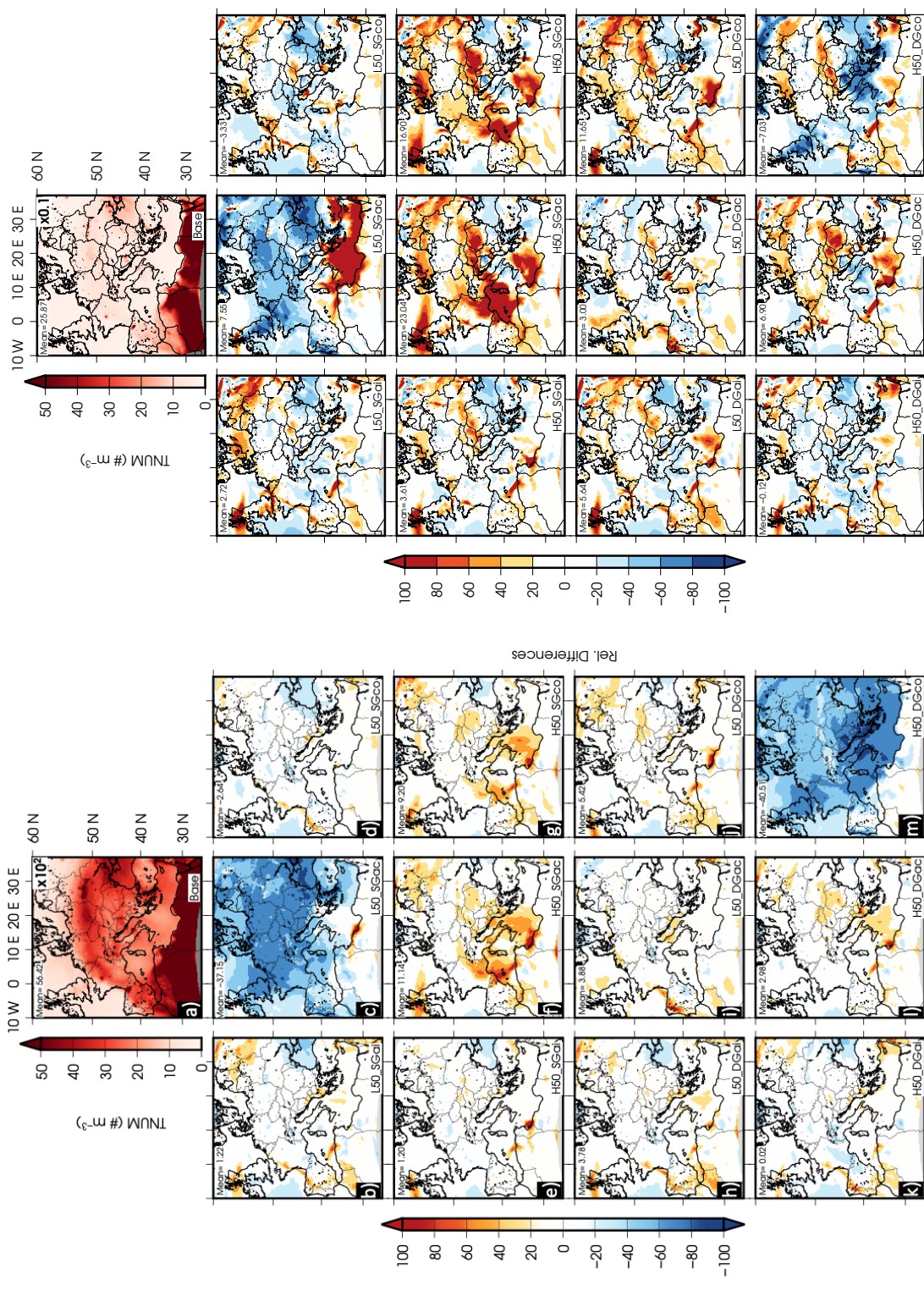

**Figure 5.** Total number concentration of particles at 1000 hPa in the Aitken and accumulation (left) and coarse (right) modes for the base case and relative differences for sensitivity test simulations at 50 %. Accumulation mode: a) Base case; b) Aitken mode 50% reduction in SG; c) accumulation mode 50% reduction in SG; d) coarse mode 50% reduction in SG; e) Aitken mode 50% increase in SG; f) accumulation mode 50% increase in SG; g) coarse mode 50% increase in SG; h) Aitken mode 50% reduction in DG; i) accumulation mode 50% reduction in DG; j) coarse mode 50% reduction in DG; k) Aitken mode 50% increase in DG; l) accumulation mode 50% increase in DG; m) coarse mode 50% increase in DG. n-z) Id. for the coarse mode.

These changes could be attributed to a narrowed distribution of the accumulation mode. This leads to an increase in the number (and mass) of particles in the coarse mode. This increase presents two different scenarios: (1) Over the central Mediterranean Sea, where fine particles dominate, the number of particles in the coarse mode increases and now this type of particles dominates, resulting in an increase of AOD since particles become larger. (2) Over the European continent, where coarse particles come predominantly from the Saharan desert dust outbreak, two aspects have to be highlighted: on the one hand, particles are removed from the accumulation mode due to a narrowed size distribution; and on the other hand, an increase in the total number concentration is expected, but this increase favors deposition and finally results in a reduction (smaller than for the accumulation mode) of the total number concentration also in the coarse mode. This preferential removal during atmospheric transport of coarse particles was previously observed by Maring et al. (2003). This reduction does not result in a significantly different PM-ratio because fine particles and coarse particles are removed, but it leads to a decrease in AOD due to a reduction in total mass concentration (see Figure 6).

These changes could also be ascribed to modifications in atmospheric transport patterns caused by ARI and ACI (taken into account in the simulations), which could alter atmospheric dynamics. However, changes in the sea level pressure (SLP, see Figure 18 in the Supplementary Material), a proxy for changes in the atmospheric transport patterns, are negligible when compared to other works that attribute changes in AOD to modifications in atmospheric dynamics (e.g. Palacios-Peña et al., 2019b).

For the experiment reducing the geometric diameter of the accumulation mode (L50_DGac), the PM-ratio as well as the total number of particles (Figure 5) and mass (Figure 6) concentrations do not show remarkable differences with respect to the base case (< 20 %). Thus, the reduction observed in AOD can be attributed to the reduction in the diameter assumed by the log-normal distribution in the accumulation mode. Hence, although mass and number concentrations are similar, the model is assuming that particles in the accumulation mode are smaller than in the base case, leading to a lower AOD.

Finally, the sensitivity experiment increasing the geometric diameter of the coarse mode (H50_DGco) leads to a general reduction of AOD, which in this case is associated to a reduction in the PM-ratio. The sum of Aitken and accumulation modes exhibits a reduction up to -60 % of the total number concentration with respect to the base case over most of the domain. However, for the coarse mode, the total number concentration remains roughly constant. Thus, the reduction in AOD comes from the decrease in the total number concentration in the Aitken and accumulation modes. It should be highlighted that while in the coarse mode the total number concentration remains constant, the total mass concentration increases (>80 % with respect to the base case over certain areas) likely because particles with a higher diameter are considered. Similar results were found by Porter and Clarke (1997), whose data demonstrated that both the accumulation and coarse mode particles gradually shifted to larger diameters as the aerosol mass increased. The reduction of mass and number in the Aitken and accumulation modes comes from a redistribution through the total size distribution caused by the increase in the coarse diameter, which produces a relocation of number and mass particles from the finer modes to the coarser.

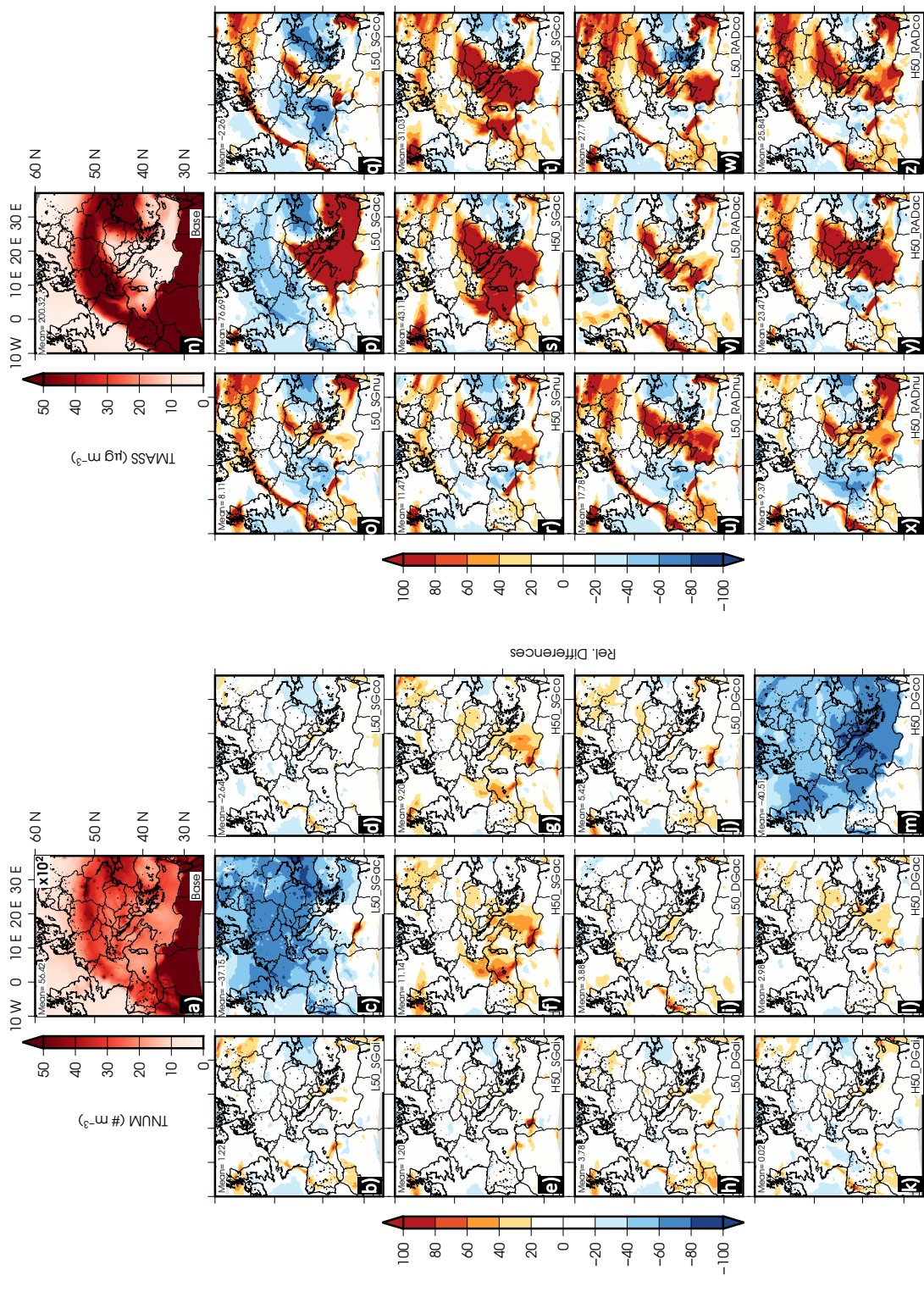

**Figure 6.** Id. 5 but for total mass concentration. Accumulation mode: a) Base case; b) Aitken mode 50% reduction in SG; c) accumulation mode 50% reduction in SG; d) coarse mode 50% reduction in SG; e) Aitken mode 50% increase in SG; f) accumulation mode 50% increase in SG; g) coarse mode 50% increase in SG; h) Aitken mode 50% reduction in DG; i) accumulation mode 50% reduction in DG; j) coarse mode 50% reduction in DG; k) Aitken mode 50% increase in DG; l) accumulation mode 50% increase in DG; m) coarse mode 50% increase in DG. n-z) Id. for the coarse mode.

## 3.4 Discussion: uncertainties in DG and SG regarding observations of aerosol size distributions

A question arising from the results presented so far relies not only on which modification presents the highest sensitivity for modifying AOD, but also how the modifications implemented in the GOCART aerosol scheme (which assumes the fixed size distribution defined in Table 2 for each experiment) compare with observations. In this sense, this section tries to bring some light on the relationship between the findings presented here and observed aerosol size distributions available in the scientific literature. To cope with that, Table 3 summarizes the observed DG and SG found through a comprehensive literature review, and selecting those works using the same definition of the log-normal function as that described in Equation 1.

**Table 2.** DG and SG values in our experiments (DG in $\mu$m).

|       | Aitken | | Accumulation | | Coarse | |
| --- | --- | --- | --- | --- | --- | --- |
|       | DG | SG | DG | SG | DG | SG |
| Base | 0.010 | 1.70 | 0.070 | 2.00 | 1.0 | 2.50 |
| L10 | 0.009 | 1.53 | 0.063 | 1.80 | 0.9 | 2.25 |
| H10 | 0.011 | 1.87 | 0.077 | 2.20 | 1.1 | 2.75 |
| L20 | 0.008 | 1.36 | 0.056 | 1.60 | 0.8 | 2.00 |
| H20 | 0.012 | 2.04 | 0.084 | 2.40 | 1.2 | 3.00 |
| L50 | 0.005 | 0.85 | 0.035 | 1.00 | 0.5 | 1.25 |
| H50 | 0.015 | 2.55 | 0.105 | 3.00 | 1.5 | 3.75 |

Table 3 is representative of the large uncertainty existing when characterizing the different modes of aerosol distributions. These values have been derived from a wide range of environments and over different locations worldwide. Regarding the geometric diameter, none of the works reviewed displays a three-mode size distribution analogous to the parameters used in GOCART (base case). However, some similarities can be found. Regarding the smallest particles, the GOCART model represents an only mode (Aitken), whose values are similar to those modes called by Covert et al. (1996) as *ultrafine*; or by Mäkelä et al. (2000) or by Rissler et al. (2006) as *nucleation*. Vakkari et al. (2013) found similar but higher values for a ultrafine/nucleation mode. Thus, the experiment in which DG increases (H10, H20 and H50_DGai) will a priori represent better this mode. However, even the H50_DGai experiment displays lower values (0.015) than those found by Tunved et al. (2003) (0.0294 and 0.0308) in the boundary layer over the Scandinavian Peninsula. So GOCART model seems to be underestimating the DG for the Aitken mode over the target domain, and the H50_DGai could contribute to enhance the skills of the modeling results.

The so-called Aitken mode by Covert et al. (1996); Mäkelä et al. (2000); Tunved et al. (2003); Rissler et al. (2006) and Brock et al. (2011) shows a similar value to the mode 2 in Porter and Clarke (1997); Vakkari et al. (2013) and Marinescu et al. (2019). These values are slightly smaller or in the range of the mode named *accumulation* in the GOCART model. Again, the cases in which DG for the accumulation mode is increased are expected to improve the representation of this mode. Moreover, the

decrease in the DG for the accumulation mode is one of the cases with remarkable differences for AOD. Thus, special attention should be paid for a correct definition of this mode.

**Table 3.** Summary of published observed log-normal size distribution parameters.

| Reference | Location | Measurement range | Environment | Mode* | Range ($\mu$m) | DG ($\mu$m) | SG |
|---|---|---|---|---|---|---|---|
| Whitby et al. (1972) | Pasadena (CA,US) | 0.003–6.8 | smog | ac | < 1 | 0.302 | 2.25 |
| | | | aerosol | co | 1 -15 | 7–10 | NA |
| Whitby (1978) | – | Review | – | nu | NA | 0.015-0.04 | 1.6 |
| | | | | ac | NA | 0.15-0.5 | 1.6-2.2 |
| | | | | co | NA | 5-30 | 2-3 |
| Covert et al. (1996) | Artic Ocean | 0.003-0.5 | Marine BL[1] | ul* | NA | 0.014±0.00042 | 1.36±0.50 |
| | | | | ai | NA | 0.045±0.00033 | 1.50±0.44 |
| | | | | ac | NA | 0.171±0.00027 | 1.64±0.25 |
| [†] Porter and Clarke (1997) | Pacific and Indian | 0.02-7.5 | Marine BL and FT[2] | 1 | NA | 0.179 | 1.46 |
| | | | | 2 | NA | 0.0765 | 1.61 |
| Mäkelä et al. (2000) | Hyytiälä (Finland) | 0.003-0.5 | boreal forest | nu | NA | 0.01548 | 1.47 |
| | | | | ai | NA | 0.05228 | 1.53 |
| | | | | ac | NA | 0.2039 | 1.40 |
| Deshler et al. (2003) | Stratocpheric (20km) Wyoming USA | 0.15-2 | Volcanic | 1 | NA | 0.13 | 1.26 |
| | | | | 2 | NA | 0.41 | 1.30 |
| | | | Background | 1 | NA | 0.69 | 1.63 |
| | | | | 2 | NA | 0.42 | 1.11 |
| [‡] Tunved et al. (2003) | Scandinavian Peninsula | 0.003-0.9 | BL(Winter) | nu | <0.03 | 0.0294 | 1.72 |
| | | | | ai | 0.03-0.11 | 0.0643 | 1.65 |
| | | | | ac | 0.11-1 | 0.198 | 1.50 |
| | | | BL(Summer) | nu | <0.03 | 0.0308 | 1.63 |
| | | | | ai | 0.03-0.11 | 0.0649 | 1.55 |
| | | | | ac | 0.11-1 | 0.187 | 1.17 |
| Rissler et al. (2006) | Rondonia[d] | 0.003-3.3 | BB plumes | nu | NA | 0.012 | 2.00-2.13 |
| | | | | ai | NA | 0.061-0.092 | 1.50-1.74 |
| | | | | ac | NA | 0.128-0.190 | 1.48-1.55 |
| Petzold et al. (2007) | European west coast | 0.004-20 | Transported BB plume | ac | NA | 0.25-0.3 | 1.30 |

**Table 3.** (Continued) Summary of published observed log-normal size distribution parameters.

| Reference | Location | Measurement range | Environment | Mode* | Range (μm) | DG (μm) | SG |
|-----------|----------|-------------------|-------------|-------|------------|---------|-----|
| Brock et al. (2011) | Denver, Florida, Alaska, and Artic | 0.004 - 8.3 | Sea-ice BL | ac | NA | 0.178 | 1.52 |
| | | | FT background haze | ai | NA | 0.008-0.05 | NA |
| | | | | ac | NA | 0.17 | 1.54 |
| | | | | co | 1 - 5 | NA | NA |
| | | | Anthropogenic plumes | ac | NA | 0.174 | 1.54 |
| | | | BB plumes | ac | NA | 0.189 | 1.50 |
| | | | | co | NA | ∼4 | NA |
| Vakkari et al. (2013) | Botsalano[e] | 0.012-0.84 | Anthropogenic plumes | 1 | NA | 0.0181 | 2.02 |
| | | | | 2 | NA | 0.0602 | 2.00 |
| | | | | 3 | NA | 0.185 | 1.39 |
| | Marikana[e] | | | NA | 1 | 0.0129 | 1.87 |
| | | | | 2 | NA | 0.0535 | 2.07 |
| | | | | 3 | NA | 0.2056 | 1.30 |
| Brock et al. (2016) | South-East US | 0.004-1.0 | summertime lower troposphere | NA | NA | 0.12-0.17 | 1.42-1.60 |
| Marinescu et al. (2019) | Southern Great Plains, USA | 0.007-14 | Rural, continental site | 1 | NA | 0.0053 | 2.80 |
| | | | | 2 | NA | 0.05866 | 1.82 |
| | | | | 3 | NA | 0.16624 | 1.53 |
| | | | | 4 | NA | 0.82355 | 1.97 |

†Mean of 9 cases; ‡Winter=Sep-Feb mean, Summer=Mar-Aug mean over different locations

[a] Niger; [b] Cape Verde; [c] Israel; [d] Amazon region; [e] South Africa

[1] BL=Boundary layer; [2] FT=Free troposphere; [3] BB=Biomass Burning

*nu=nucleation, ul=ultrafine

Finally, the model used in this contribution considers a coarse mode with DG 1$\mu$m. Again, this value might be underestimated because the literature reviewed (Table 3) found DG for the coarse mode with values higher than 2$\mu$m (maximum value of 1.5$\mu$m in our H50 experiment) and up to 30. Thus, particles in our model are modelled smaller than those generally observed. Moreover, the increase of the DG in the coarse mode is one of the case in which AOD shows noticeable differences. Marinescu et al. (2019) found a mode number 4 with DG lower but close to the coarse mode in our simulations. Henceforth, increasing the DG in the coarse mode in GOCART model could improve the results of AOD in our simulations.

Values taken by SG in our base case are similar to those reported by Whitby (1978). However, observed SG are highly uncertain. Most of these works found SG values lower than those used in our base case for all of the modes: (1) ultrafine/nucleation, which corresponds with our Aitken; and (2) Aitken and accumulation, which are represented by our accumulation and coarse modes. Tunved et al. (2003) observed SG values similar to the ones used for the Aitken mode in the base case (1.70), both in winter and summer in the boundary layer over the Scandinavian Peninsula. However, this value is underestimated in comparison with the measurements carried out by Rissler et al. (2006); Vakkari et al. (2013) and Marinescu et al. (2019).

Whitby et al. (1972) is the only work in which SG value is higher than 2 for the accumulation mode. The rest of works reported lower values than the value used by GOCART in the base case. Probably because of that, the reduction of this parameter is the case which shows a higher influence in AOD representation (for all the experiments 10, 20 and 50%). As also happened for the accumulation mode, the SG in the base case for the coarse mode is highly overpredicted by the model when comparing its value with observations available in the literature.

## 4    Summary and Conclusions

Aerosol size distribution is, among others, a key property of atmospheric aerosols which importantly determines the aerosol interactions with radiation and clouds, since optical properties (e.g. AOD) strongly depend on aerosol size distribution. Moreover, this distribution exerts a strong influence on ARI and its associated radiative forcing. Henceforth, the main objective of this contribution is to study the impact of the representation of aerosol size distribution on aerosol optical properties over central Europe, and particularly over the Mediterranean Basin during summertime. The case study has been selected because the Mediterranean Basin presents an intense formation, accumulation and recirculation of aerosols from different sources, intensified during this summer episode.

In order to fulfill the objectives, a sensitivity test has been carried out by perturbing the parameters defining a log-normal size distribution ($\pm$10, 20 and 50 %). The sensitivity experiments reveal that modifying (lowering) the standard deviation of the accumulation mode (L_SGac) presents the highest sensitivity with respect to the AOD representation. This modification provokes a narrowed distribution in the accumulation mode resulting in two different scenarios: (1) over those areas where fire particles predominate in the base case, the transfer of particles from the accumulation to the coarse mode results in an increase of the total number and mass in the latter mode and an increase in AOD; and (2) over those areas where coarse particles dominate, particles are transferred from the accumulation to the coarse mode albeit this favors the removal of particles, reducing

the total number and mass and hence the levels of AOD. This removal of particles of the coarse mode during atmospheric transport was previously observed by Maring et al. (2003).

The reduction of the standard deviation of the accumulation mode is the only experiment in which all of the sensitivities tests run present important influences on AOD. Moreover, the response for all of the sensitivities is similar and increases as the

modification becomes larger.

The rest of the sensitivity experiments only show significant differences when modifying the target parameters by 50 %. The experiment in which the diameter of the coarse mode is increased (H50_DGco) exerts the largest influence on AOD levels. For this experiment, a redistribution through the total size distribution occurs due to the increase in the coarse diameter, which produces a relocation of number and mass particles from the finer modes to the coarse. The other experiment showing

an important response to the perturbations of sensitivity test is the case in which the diameter of the accumulation mode is decreased (L50_DGac). Here, the reduction observed in AOD could be attributed to the reduction of the diameter assumed by the log-normal distribution of the accumulation mode. Hence, although mass and number concentrations are similar, the model is assuming that particles in the accumulation mode are smaller than in the base case, leading to a reduction of AOD.

The comparison of size distribution parameters (DG and SG) in the simulations and observations reveals that, generally, the

base case underestimates the geometric diameter in all modes. This underestimation is even more noticeable for the coarse mode. Moreover, a mode is missed for the fine particles. While the model includes two modes (Aitken and accumulation) for particles lower than $1\mu$m, observations indicate the presence of three modes (ultrafine/nucleation, Aitken and accumulation). The differences found in the experiments when the DG is modified in the accumulation and coarse mode evince the need to carefully consider the definition of the value of this parameter in GOCART.

On the other hand, the modifications made to the standard deviation of the accumulation mode in the sensitivity experiments highly influence the AOD levels. This fact, together with the high uncertainty when measuring this parameter (as reported by observations) should be taken into account in order to improve the representation of size distribution in aerosol models (in particular, in those using a fix size distribution as GOCART).

This contribution identifies those cases where AOD exhibits a higher sensitivity to the target parameters. However, further

experiments are needed in order to improve the representation of size distribution in models by using observational data (information for DG and SG from in-situ and remote sensing observations). Although a more accurate fixed size distribution could be defined, the use of any fixed distribution has some limitations since aerosol size varies in space and time. The improvement in this representation will reduce the uncertainty associated to the effects of aerosols on climate, in particular related to ARI.

*Code and data availability.* WRF-Chem code used to perform this work as well as data presented here are available at doi:10.5281/zenodo. 3768076

*Author contributions.* LP-P wrote the manuscript, with contributions from PJ-G. LP-P, PJ-G and JF design the experiments; LP-P conducted the numerical simulations and compiled all the experiments. LP-P did the analysis, with the support of JF , EP-S and PJ-G.

*Competing interests.* The authors declare no conflict of interest.

*Acknowledgements.* The authors are thankful to the WRF-Chem development community and the G-MAR research group at the University of Murcia for the fruitful scientific discussions, especially to Dr. Juan Pedro Montávez.

*Financial support.* This study was supported by the Spanish Ministry of the Economy and Competitiveness/Agencia Estatal de Investigación and the European Regional Development Fund (ERDF/FEDER) through project ACEX-CGL2017-87921-R project. L. P.-P. was supported by the FPU14/05505 grant from the Spanish Ministry of Education, Culture and Sports and the ERASMUS+ program. Jerome D. Fast
was supported by the Atmospheric System Research (ASR) program as part of the U.S. Department of Energy's Office of Biological and Environmental Research.

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
