# Peer review of "Sensitivity of aerosol optical properties to the aerosol size distribution over central Europe and the Mediterranean Basin"

_Geoscientific Model Development, 2020_

## Referee Comment (RC1) · Anonymous Referee #1 · 15 Jul 2020

The paper presents the results of unpublished research concerning a sensitivity study of aerosol optical properties to the aerosol size distribution over central Europe and the Mediterranean Basin. Methodology and the graphical representation (figure 1-5) of this work is not described properly and in some part the text is quite obscure. In these conditions, I think there are not the ideal conditions to publish this work so I recommend Major revision.

Below a list of some more specific questions

Abstract Please avoid use of acronyms in the abstract, there are many: ARI-ACI-AOD-DG-SG

Introduction In my opinion lines 25-44 are not appropriate for the introduction, I would rather move lines 25-44 to another section (methodology ???). As a consequence, introduction should be rewritten considering recent papers on this specific subject. The application of the GOCART aerosol scheme of the WRF-Chem model should be also mentioned and referenced. See for example: Geosci. Model Dev., 12, 131–166, 2019 https://doi.org/10.5194/gmd-12-131-2019

2 Methodology This chapter should include one more paragraph in which the synoptic analysis is reported including figures from possibly ERA5 or GFS. What is written in lines 63-66 is not sufficient to describe the synoptic conditions and quite superficial.

2.1 Model setup Lines 77-79: To my knowledge only ARI are considered in WRF-Chem with GOCART aerosols (using RRTMG radiation modules). If you want to consider Aerosol Clouds Interaction you should use more complex aerosol representation like MOSAIC or MADE/SORGAM for example.

In lines 80-81 authors write: "The case study selected here trusts on an extended episode evaluated in Palacios-PenÌČa et al. (2019b). The model setup for all the experiments is the same as for that work. However, a brief summary of the configuration is included here" In my opinion each paper should be self-sufficient and not relying in other works even if from the same authors. In this context I think it would be necessary to report the physics and chemistry setup in a table and describe also which option is used for chem_opt (301 ???) and for dust_opt (1 ????).

Line 99: pls show the domain(s) with an additional figure.

Lines 105-107. To my knowledge the GOCART aerosol schemes consider a bulk formulation for BC, OC and sulfate but it is sectional for mineral dust using Kok (2011) brittle fragmentation theory to obtain (dust_1, dust_2, dust_3, dust_4, dust_5) variables. So GOCART is bulk/sectional and not modal like described here. Please clarify this point and explain also the following (lines 106/107): "The selection of this scheme is conditioned by the fact that WRF-Chem version 3.9.1.1 only allows the simulation of

desert dust and sea salt with this GOCART scheme"

**3.1 Effects on AOD representation**

Fig.1 is quite difficult to interpret. The panels are described by a series of acronyms that are not introduced before, the legend is not descriptive at all. As a consequence, it is really difficult to understand what it is written in the text (lines 128-). Another question concerns the top-row in fig.1 which should represent the baseline case, that is "the temporal mean of AOD for the target period". Then, both "temporal mean" and "target period" should be defined. A comparison with MODIS (????) should be introduced and showed for the same target period. My personal analysis of MODIS combined DT and DB for both Aqua and Terra spacecrafts show a quite different AOD distribution in the target period. So I would suggest to add HERE the comparison with experimental data (AOD/MAIAC ???) of the baseline case.

Line 137 – abroad => onboard

Line 147 – "Sensitivity experiments regarding modifications in the accumulation mode lead to higher changes with respect to the Aitken mode" where is this showed ????

**3.2 Significance of AOD changes**

Line 167 – "(in space and time)" what does it means ????

Line 170 – The Kolmogorov-Smirnov test is not introduced and/or explained.

Line 171-172 : "This test estimates the distance between the cumulative distribution function (represented by D) and how significant this difference is (represented by the p.value). " D and p.values has not been introduced

Fig.2 What is reported in the abscissa ???? panels should be numbered and described in the legend together with the meaning of the red dotted line and black continuous line

Overall this paragraph is of difficult interpretation, KS test is not described, fig.2 is not properly defined and plagued by hundreds of acronyms. Please organize this para-

Interactive
comment

graph in a more readable way at least introducing tables with results, in order to help reader in the interpretation of what is written here.

3.3 Disentangling the causes of AOD variations due to size distribution

How PM (2.5 and 10) are calculated ??? I mean for the optical properties you introduced aitken-fine-coarse mode, and whart happen for the the mass ???? WRF-Chem has specialized routines to calculate PM from GOCART aerosol mechanism which is not modal.

Figure 3. Please explain why there is such abrupt change of PM25/PM10 ratio following the African coastline (top row) and why this ratio is almost zero in Africa.

This predominance of fine vs coarse particles or viceversa may be verified against MODIS products, see for example the ang exp and/or maiac algorithm.

Line 205

Figures 4 and 5 Authors should find a way to better represent their work, figures are turned 90° left with the legends not properly exhaustive
* * *

---

## Referee Comment (RC2) · Anonymous Referee #2 · 31 Aug 2020

This manuscript summarizes an interesting and potentially useful study to examine the effect of varying the aerosol size distribution parameters within the WRF-Chem model on aerosol-radiation interactions. This work is an examination of findings in the literature that aerosol optical depth and radiative effects are sensitive to modal parameters such as geometric mean diameter and standard deviation, and that descriptions in models need to more closely align with observational evidence for narrower size distributions than are typically prescribed.

I am an experimentalist, so my understanding of the use of modal representations of the aerosol size distribution in the WRF-Chem model is limited. From the manuscript,

the model uses the GOCART scheme, in which (lines 108-111) " bulk mass and number is converted into an assumed log-normal modal distribution, then dividing the mass into sections or bins." So from this I gather that the mass for different aerosol components is carried in the model, and then different components are divided into different lognormal distributions (e.g., dust and sea-salt mass would go in the coarse lognormal mode, sulfate into the accumulation and Aitken lognormal modes, etc.). These lognormal distributions have prescribed geometric mean diameter Dg and geometric standard deviation Sg, while the mass varies with the abundance of each component as predicted by GOCART. The number follows from the mass; it is not explicitly predicted by GOCART. Once this mass apportionment has been done, the lognormal distributions are chopped into discrete bins, the number of particles in each bin is calculated, optical and hygroscopic properties for each component are assigned, and then the Mie parameterization is applied to calculate optical parameters such as extinction and absorption.

Assuming this is correct, I see some substantial issues with the manuscript as currently written. My major concerns are as follows: 1) There seems to be a fundamental lack of understanding of the lognormal function. For example, lines 43-44, it is stated that the 3rd moment is mass and the 4th is volume. In fact, mass is just the 3rd moment (volume), multiplied by particle density. Furthermore, there are (at least) two variations of the lognormal function found in the aerosol literature. One is:

F=(N*ln(10)/(sqrt(2*pi)*ln(Sg)))*exp(-.5*(ln(D/Dg)/ln(Sg))^2), where N is the particle number (or volume if the lognormal describes the volume distribution). This is the differential function, describing dN/dlogDp.

The other common formulation is:

F=N*exp(-1*(ln(D/Dg)/ln(Sg))^2)

In the one I'm most used to, the first, the geometric standard deviation represents the multiplicative standard deviation. For example, for a lognormal distribution with a Dg

of 1 $\mu$m and a Sg of 1.5, 63% of the distribution would lie with diameters between 1/1.5 and 1*1.5 $\mu$m. In this formulation, Sg can never be <1; a Sg of 1 represents a monomodal aerosol. This is the formulation that is used in most, but not all, of the values for Sg in Tables 2 and 3. In the second formulation, I believe Sg can be <1. This is likely the formulation used in Remer et al., 1998, listed in Table 2.

For the current manuscript the values for Sg used for the base case are 1.7, 2.0, and 2.5 for the Aitken, accumulation, and coarse modes, which would imply the first formulation was used. However, if this is the case, the sensitivity case for L50_Sgai is run with a value of 0.85, which is physically impossible. The sensitivity case for L50_Sgac has a value of 1.00, which would imply a monomodal size distribution.

2) The values of Dg for the accumulation mode listed in Table 1, "found through a comprehensive review", are implausibly small. The accumulation mode is often defined as having diameters from 0.1 to 1.0 $\mu$m, yet all the values except the "H50" in this table list a Dg of <0.084 $\mu$m. I have a strong suspicion that radius has been confused for diameter.

3) At lines 225-227, the case of "L50_SGacc", which I assume means a reduction in 50% in the geometric standard deviation of the accumulation mode relative to the base case, is described as leading to a reduction in both number concentration and in mass. This is nonsensical; reduction of the standard deviation of a lognormal distribution does not change the integrated amount of the parameter; it merely changes the width of the PDF. If the lognormal is describing the distribution of mass in the mapping of GOCART mass to diameter, then the mass should be held constant as the standard deviation changes (number would change, however). Alternatively, if the number is held constant, then the mass will change. But both should not change if just the standard deviation is changed.

There are a number of more minor issues with the manuscript, including no description of the coding used for the different cases ("L50_Sgac", "H20_Dgai", etc.). Figure captions do not describe the contents of the figures (for example, in Figure 2 there is no explanation in the caption for the curves that are plotted.) And there are many unexplained parameters and typographical errors. I recommend a thorough proofreading.

To be acceptable for publication, I recommend the following: 1) Clearly define the lognormal function that is being used. Please read J. Heintzenberg, "Properties of the Log-Normal Particle Size Distribution" in Aerosol Science and Technology, 1994, https://doi.org/10.1080/02786829408959695. Also, the Seinfeld and Pandis textbook has a substantial section on the lognormal distribution that should be reviewed.

2) Clearly explain how the GOCART masses are mapped to the lognormal functions (i.e., is mass being conserved during the sensitivity tests, or is mass?). How are the different components–dust, sea-salt, secondary sulfate–being mapped to the different lognormal modes? Probably a separate section with equations and a thorough verbal description of this mapping is warranted.

3) Ensure that the range of literature values are all defined using the same definition of the lognormal function. My preference would be to use the more common definition listed above, wherein all values of Sg are >1.

4) Make sure that diameter, rather than radius, is used consistently.

5) Make sure all figure captions describe the contents of the figures clearly. For example, Fig. 1 caption currently says, "AOD at 550nm and differences for simulations of sensitivity test at 50%". It should say something like, "Map showing the difference between the base case and sensitivity tests using 50% changes in parameters. a) Base case showing AOD. b) Aitken mode with 50% reduction in Sg. c) Accumulation mode with 50% reduction in Sg." etc. Or have much larger, clearly defined labels on the columns and rows.

6) Perform a very thorough proofreading, making sure all sentences are logical and complete and looking for grammatical errors. Perhaps Dr. Fast could help with this.

[Figure]

It is a very useful exercise to investigate how uncertainties in assumptions about aerosol size distributions are reflected in AOD, and this manuscript has good potential. However, the fundamental confusion about the definition of the lognormal function being used and the appropriate range of lognormal parameters for sensitivity tests makes me concerned that there are substantial errors embedded in the calculations and results. These, and issues of presentation, must be dealt with before the manuscript can be considered again for publication in GMD.

---

## Author Comment (AC1) · 28 Sep 2020

*Dear Editor, Geoscientific Model Development Discussion:*

*Please find below our item-by-item response to the Reviewer's comments regarding manuscript* **"Sensitivity of aerosol optical properties to the aerosol size distribution over central Europe and the Mediterranean Basin"** *by L. Palacios-Peña et al.*
*Do not hesitate to contact us with further questions.*

*With kind regards,*

*Laura Palacios Peña*

*First of all, we would gratefully thank the Editor and Reviewers for their valuable comments, leading to a noticeable improvement of the manuscript.*

Anonymous Referee #1:

*Q: Abstract Please avoid use of acronyms in the abstract, there are many: ARI-ACI-AODDG-SG*

*A: The abstract has been reviewed in order to reduce as possible the use of acronyms.*

*Q: Introduction In my opinion lines 25-44 are not appropriate for the introduction, I would rather move lines 25-44 to another section (methodology ???). As a consequence, introduction should be rewritten considering recent papers on this specific subject. The application of the GOCART aerosol scheme of the WRF-Chem model should be also mentioned and referenced. See for example: Geosci. Model Dev., 12, 131–166, 2019https://doi.org/10.5194/gmd-12-131-2019*

*A: Following the reviewer's suggestion, this part of the introduction has been moved to the methodology section and we have added the following information in the Introduction:*

*"In this sense, the representation of aerosol processes in meteorological or climate models presents a high uncertainty (Boucher et al., 2013). Particularly, modelling aerosol size distribution introduces a noticeable uncertainty in chemistry transport models (Tegen and Lacis, 1996; Claquin et al., 1998). Three different approaches are usually employed for aerosol models: 1) the bulk approach, in which only the aerosol mass concentration is computed; 2) the modal approach, which uses multiple superposed modes; and 3) the sectional representation, which discretizes the aerosol size distribution into classes or bins. These three approaches are deeply described in the Methodology section.*

*These three approaches for aerosol representation are included in the WRF-Chem model, which is the coupled chemistry-meteorological model used in this work. The sectional approach is used by the Model for Simulating Aerosol Interactions and Chemistry (MOSAIC; Zaveri and Peters, 1999) and a simple scheme for volcanic ash (Stuefer et al., 2013). With respect to the modal approach, the schemes available within WRF-Chem are the Modal Aerosol Dynamics Model for Europe (MADE; Ackermann et al., 1998) and the Modal Aerosol Model from CAM5 (MAM; Liu et al., 2012). Finally, the Goddard Global Ozone Chemistry Aerosol Radiation and Transport (GOCART; Ginoux et al., 2001; Chin et al., 2002) uses the bulk approach.*

*Some of these schemes have been widely applied for the study of aerosol optical properties and their uncertainty. In this sense, the evaluation of aerosol optical properties as represented by the MOSAIC has been conducted by Barnard et al. (2010) or Lennartson et al. (2018) to analyze the diurnal variation of AOD. Chapman et al. (2009) went a step beyond and evaluated the radiative impact of including coupled aerosol-cloud-radiation processes. In addition, some contributions had the objective of assessing the representation of aerosol optical properties and their uncertainties using MOSAIC together with other schemes, mainly MADE (Zhao et al., 2010, 2011, 2013; Balzarini et al., 2015; Yang et al., 2018; Saide et al., 2020). The GOCART scheme has also been used for this aim. For example, LeGrand et al. (2019) compared the Air Force Weather Agency (AFWA) dust emission scheme withing GOCART to other dust emission schemes available in WRF-Chem and their skills for representing AOD. In this former work, the need for tuning the model in order to get a reasonable simulation of AOD for each location and/or event was pointed out based on the results of Bian et al. (2011); Dipu et al. (2013); Kumar et al. (2014); Jish Prakash et al. (2015); Zhang et al. (2015); Kalenderski and Stenchikov (2016); Hu et al. (2020); among others. All those works evaluated the representation of AOD depending on the approach followed for the aerosol scheme. However, this contribution evaluates the uncertainty associated to the representation of the aerosol size distribution when estimating aerosol optical properties.".*

*Q: 2 Methodology This chapter should include one more paragraph in which the synoptic analysis is reported including figures from possibly ERA5 or GFS. What is written in lines 63-66 is not sufficient to describe the synoptic conditions and quite superficial.*

*A: We agree with the reviewer's comment. Henceforth, a paragraph describing the synoptic situation has been included. A new Figure has been added to the Supplementary Material, adapted from Palacios-Peña et al. (2019b), where a detailed description of the episode is presented. Hence, the following text has been added to the revised version of the manuscript:*

*"The case study selected here covers an extended episode between 4 and 9 of July, 2015. The synoptic description of the synoptics conditions has been widely presented in Palacios-Peña et al. (2019). Nonetheless, a brief summary of the meteorological episode is presented here.*

*During this episode, the development of an omega-blocking situation takes place, with low pressure over western England. The episode presents a high stability over the Mediterranean Basin with a high aerosol load, fire emissions in the target area, and a strong dust outbreak induced by the penetration of warm air and dust from northwestern Africa toward western Mediterranean Sea and northern Europe (Nabat et al., 2015). The weakening of this synoptic situation results in a cyclonic circulation of the air over the western Mediterranean (Palacios-Peña et. al, 2019b).*

*The choice of this episode reveals the crucial role of aerosols from different sources over the Mediterranean Basin, whose forcing is even stronger in summertime."*

*Q: 2.1 Model setup Lines 77-79: To my knowledge only ARI are considered in WRF-Chem with GOCART aerosols (using RRTMG radiation modules). If you want to consider Aerosol Clouds Interaction you should use more complex aerosol representation like MOSAIC or MADE/SORGAM for example.*

*A: The reviewer's comment raises an interesting point. While a full aerosol-cloud coupling is not allowed in WRF-Chem when GOCART is used, the conversion of a single moment microphysics into a double moment is allowed with greater flexibility in representing size distributions and hence microphysical process rates. The next paragraph has been included in the manuscript in order to clarify this fact:*

*"The GOCART aerosol scheme in WRF-Chem does not allow a full coupling of aerosol-cloud interactions. For instance, convective wet scavenging (conv_tr_wetscav), in-cloud wet scavenging and cloud chemistry are not available. However, in those simulations denoted as ACI, the Morrison microphysics (used in this contribution) acts as a double-moment scheme meanwhile in the rest of the simulations it works as a single-moment microphysics scheme. This latter approach is unsuitable for assessing aerosol-clouds interactions because it does not represent a prognostic treatment of droplet number. Hence, ACI configuration allows a double-moment microphysics with greater flexibility when representing size distributions and hence microphysical process rates (Palacios-Peña et al., 2020). When the double-moment scheme is activated, a prognostic droplet number concentration using gamma functions and mixing ratios of cloud ice, rain, snow, graupel/hail, cloud*

*droplets and water vapour are estimated (Morrison et al., 2009). Finally, the interaction of cloud and solar radiation with Morrison microphysics is implemented in WRF-Chem (Skamarock et al., 2008). Therefore, droplet number will affect both the calculated droplet mean radius and cloud optical depth."*

*Q: In lines 80-81 authors write: "The case study selected here trusts on an extended episode evaluated in Palacios-Peña et al. (2019b). The model setup for all the experiments is the same as for that work. However, a brief summary of the configuration is included here" In my opinion each paper should be self-sufficient and not relying in other works even if from the same authors. In this context I think it would be necessary to report the physics and chemistry setup in a table and describe also which option is used for chem_opt (301 ???) and for dust_opt (1 ????).*

*A: We thank the reviewer for his/her useful suggestion. A table summarizing the model setup has been included in the revised version of the manuscript. The chem_opt is 301, as indicated in the table for the gas-phase and aerosol scheme (GOCART coupled with RACM-KPP). The dust_opt is 1; this information has been included in the table.*

*Q: Line 99: pls show the domain(s) with an additional figure.*

*A: The figure required has been included in the revised version of the manuscript.*

*Q: Lines 105-107. To my knowledge the GOCART aerosol schemes consider a bulk formulation for BC, OC and sulfate but it is sectional for mineral dust using Kok (2011) brittle fragmentation theory to obtain (dust_1, dust_2, dust_3, dust_4, dust_5) variables. So GOCART is bulk/sectional and not modal like described here. Please clarify this point and explain also the following (lines 106/107): "The selection of this scheme is conditioned by the fact that WRF-Chem version 3.9.1.1 only allows the simulation of desert dust and sea salt with this GOCART scheme"*

*A: The information included in these lines has been updated following the reviewer's comment.*

*"As aforementioned, the aerosol scheme used in the simulations is GOCART (Ginoux et al., 2001; Chin et al., 2002), which includes a bulk approach for black carbon (BC), organic carbon (OC) and sulfate; and a sectional scheme for mineral dust and sea salt using Kok (2011) brittle fragmentation theory. This is a simple and cheap computational approach. The selection of this scheme is conditioned*

*by the fact that WRF-Chem version 3.9.1.1 only allows the simulation of desert dust and sea salt with this GOCART scheme."*

*The comment on the modal distribution in the previous version of the manuscript refers not to the calculation of air concentrations, but to the estimation of aerosol optical properties, as was explained a few lines below in the manuscript. Once GOCART has estimated the bulk mass and number, this information is passed to the optical module and converted into an assumed log-normal modal distribution (distributing the bulk mass) and then dividing the mass into sections or bins ("i").*

*The comment regarding the fact that the version 3.9.1.1 only allows the simulation of desert dust and sea salt with the GOCART scheme intends to clarify that dust and sea salt emission fluxes are only available with GOCART. The inclusion of sea salt or dust in modal/sectional aerosol schemes (e.g. MADE and MOSAIC) was disabled in that version of WRF-Chem because of errors/bugs in the code of the model.*

Q: Fig.1 is quite difficult to interpret. The panels are described by a series of acronyms that are not introduced before, the legend is not descriptive at all. As a consequence, it is really difficult to understand what it is written in the text (lines 128-). Another question concerns the top-row in fig.1 which should represent the baseline case, that is "the temporal mean of AOD for the target period". Then, both "temporal mean" and "target period" should be defined.

*A: A description of the acronyms has been included in this section in order to improve the understating of the text. Moreover, we have tried to improve the legend of the figure.*

*"As aforementioned, the experiment of the sensitivity test consists in the modification of the geometric diameter (DG) and the standard deviation (SG) by ±10, 20 and 50%. Taking into account this consideration, experiments have been named indicating the sign of the modification (L for a reduction and H for an increase) and the percentage (10, 20 or 50), the variable of the size distribution modified (SG for the standard deviation and DG for the geometric diameter) and the mode in which the modification has been conducted (ai for Aitken, ac for accumulations and co for coarse. Thus, the acronym for a experiment follows the this pattern: (L|H)(10 | 20 | 50)_(SG | DG)(ai | ac | co). For example, L50_SGai involves a reduction (L) of 50% of the standard deviation (SG) in the Aitken mode (ai).*

*Top row in Figure 1 displays the hourly mean of AOD for the target period (from 4 to 9 July) of the base case."*

*Legend Figure 1: "AOD at 550nm and differences for the sensitivity tests modifying the parameters by 50%. a) Base case; b) Aitken mode 50% reduction in SG; c) accumulation mode 50% reduction in SG; d) coarse mode 50% reduction in SG; e) Aitken mode 50% increase in SG; f) accumulation mode 50% increase in SG; g) coarse mode 50% increase in SG; h) Aitken mode 50% reduction in DG; i) accumulation mode 50% reduction in DG; j) coarse mode 50% reduction in DG; k) Aitken mode 50% increase in DG; l) accumulation mode 50% increase in DG; m) coarse mode 50% increase in DG."*

*Q: A comparison with MODIS (????) should be introduced and showed for the same target period. My personal analysis of MODIS combined DT and DB for both Aqua and Terra spacecrafts show a quite different AOD distribution in the target period. So, I would suggest to add HERE the comparison with experimental data (AOD/MAIAC ???) of the baseline case.*

*This predominance of fine vs coarse particles or viceversa may be verified against MODIS products, see for example the ang exp and/or maiac algorithm.*

*A: Regarding the comparison of the model, the model has been extensively evaluated against MODIS observations in a number of manuscript (e.g. Palacios-Peña et al., 2019b; Palacios-Peña et al., 2020). However, for the sake of clarity, the following text has been introduced in the revised version of the manuscript:*

*"Top row in Figure 1 displays the hourly mean of AOD for the target period (from 4 to 9 of July). As established by Palacios-Peña et al. (2019b), high AOD values over the western part of the Mediterranean Basin and central Europe were caused by a strong desert dust outbreak from the Sahara Desert. The eastern part of the Mediterranean AOD also presents high values because the outbreak reached that part at the end of the period. Palacios-Peña et al. (2019b) evaluated the simulation of the base case of this work against observations coming from Moderate Resolution Imaging Spectroradiometer (MODIS; and instruments onboard satellite) and the Aerosol Robotic Network (AERONET). The evaluation results (Figure 2 in the Supplementary Material) demonstrated negligible errors of the model over large areas but a underestimation of AOD over the north of Germany and the central and western Mediterranean of around -0.2 which somewhere reaches up to -0.4. In spite of this underestimation, the spatio-temporal mean bias error is -0.02 (against both, MODIS and AERONET) and the mean absolute errors is 0.16 (when assessed against MODIS) and 0.12 (when evaluated against AERONET network). Palacios-Peña et al. (2019b) also pointed out that this underestimation over the central and western Mediterranean turns into a larger overestimation when a coarser resolution is used due to a worsening in the representation of the dynamical patterns and thus, the dust transport."*

*Q: Line 137 – abroad => onboard.*

*A: Corrected*

*Q: Line 147 – "Sensitivity experiments regarding modifications in the accumulation mode lead to higher changes with respect to the Aitken mode" where is this showed????*

*A: When comparing AOD changes in the first column of panel in Figure 1 (modification in Aitken mode) and in the second column (modification in the accumulation mode), changes in most areas of the panel of the second column are in general higher and with a larger spatial extension than in the first one.; in particular, changes in the L50_SG experiment (second row, second column panel), which is one of the evaluated latter. The sentence has been modified to improve its understanding.*

*"Sensitivity experiments regarding modifications in the accumulation mode lead to higher and more extended spatial changes with respect to the Aitken mode; however, the sensitivity is still limited."*

*Q: Line 167 – "(in space and time)" what does it means ????*

*A: This means that the PDF of the AOD data has been built using spatio-temporal data. Data during the target period (from 4 to 9 of July) and from the entire domain. The sentence has been rewritten for the sake of clarity.*

*"Figure 3 displays the probability density function (PDF) of the AOD at 550nm values (for all cells and timesteps in the model) simulated by the base case (solid black line) and each of the experiments (dashed red line) in the sensitivity test."*

*Q: Line 170 – The Kolmogorov-Smirnov test is not introduced and/or explained.*

*Line 171-172: "This test estimates the distance between the cumulative distribution function (represented by D) and how significant this difference is (represented by the p.value). " D and p.values has not been introduced.*

*Fig.2 What is reported in the abscissa ???? panels should be numbered and described in the legend together with the meaning of the red dotted line and black continuous line Overall this paragraph is of difficult interpretation, KS test is not described, fig.2 is not properly defined and plagued by hundreds of acronyms. Please organize this paragraph in a more readable way at least introducing tables with results, in order to help reader in the interpretation of what is written here.*

*A: A paragraph describing the K-S test and its statistical figures has been included in this section. Moreover, figure 2 and its legend has been modified following the reviewer's suggestion.*

*"The Kolmogorov-Smirnov is a non-parametric test using for the evaluation of the statistical similarity of the distribution between two datasets. The test is based in the assumed similarity of the ECDF ('Empirical Cumulative Distribution Function') between two random samples. The maximum distance between both ECDFs, normally named as 'D', indicates how far are both distributions. In this work, the distribution of each experiment (denoted by the red dashes line) has been evaluated against the distribution of the base case (black line). The p-values represents the probability of values as extreme as those obtained for samples coming from the same distribution. Low p-values shows a low probability of error when the null hypothesis is rejected and, thus, indicates that both samples does not provide from the same distribution (Sprent and Smeeton, 2016)."*

*Figure 2 legend: "PDF of AOD values for the base case (black line) and each of the sensitivity test simulations at 50 % (dashed-red line). Values in Figures represent the results of the statistic from the Kolmogorov-Smirnov test. a) Aitken mode 50% reduction in SG; b) accumulation mode 50% reduction in SG; c) coarse mode 50% reduction in SG; d) Aitken mode 50% increase in SG; e) accumulation mode 50% increase in SG; f) coarse mode 50% increase in SG; g) Aitken mode 50% reduction in DG; h) accumulation mode 50% reduction in DG; i) coarse mode 50% reduction in DG; j) Aitken mode 50% increase in DG; k) accumulation mode 50% increase in DG; l) coarse mode 50% increase in DG."*

*Labels have been included in the abscissa axis in the manuscript and the Supplementary Material.*

Q: How PM (2.5 and 10) are calculated ??? I mean for the optical properties you introduced aitken-fine-coarse mode, and whart happen for the the mass ???? WRF-Chem has specialized routines to calculate PM from GOCART aerosol mechanism which is not modal.

*A: PM10 and PM2.5 mass are directly estimated from the default routines in GOCART, as indicated by the reviewer.*

Q: Figure 3. Please explain why there is such abrupt change of PM25/PM10 ratio following the African coastline (top row) and why this ratio is almost zero in Africa.

*A: "PM-ratio of the base case (Figure 4 a) is almost zero in Africa because during this Saharan Desert dust episode there is a predominance of coarse particles*

*(PM10) over this area. When comparing top row in Figure 4 and Figure 2 areas with high AOD levels match those areas with PM-ratio close to 0 due to the influence of desert dust. The high values of the ratio over the central Mediterranean Sea could be ascribed to the transport of this dust. At the beginning of the episode, the dust outbreak reached the central Mediterranean (coarse particles from dust were modelled here), but as the episode developed, dust -and hence, PM10 particles- move eastwards and northwards, being the PM2.5 concentrations higher over the Mediterranean Sea and the African coastline at the end of the target period."*

*This paragraph has been included in the text in order to better explain the phenomena studied here.*

Q: Figures 4 and 5 Authors should find a way to better represent their work, figures are turned 90°left with the legends not properly exhaustive

*A: The caption and legends of all of the figures has been revised and improved following the reviewer's advice including figure 4 and 5.*

Anonymous Referee #2:

Q: Clearly define the lognormal function that is being used. Please read J. Heintzenberg, "Properties of the Log-Normal Particle Size Distribution" in Aerosol Science and Technology, 1994, https://doi.org/10.1080/02786829408959695. Also, the Seinfeld and Pandis textbook has a substantial section on the lognormal distribution that should be reviewed.

*A: As the reviewer guessed, the lognormal function used is the first option (F=(N\*ln(10)/(sqrt(2\*pi)\*ln(Sg)))\*exp(-.5\*(ln(D/Dg)/ln(Sg))ˆ2)). The following paragraph has been included in the methodology section in order to clarify this point.*

*"When a size distribution is considered, a log-normal approach is typically employed in chemistry transport models because this approach fits observed aerosol size distribution reasonably well and its mathematical form is convenient for dealing with the moment distribution. In this approach, all of the moment distributions are log-normal and present the same geometric mean diameter and geometric standard deviation, parameters which determine the log-normal distribution (Hinds, 2012). One of the most common log-normal size distributions used is that described in Heintzenberg (1994) (Eq. 1), which has*

*algo been employed in this contribution. Equation 1 represents the zeroth moment of the particle size distribution, where $d_{p}$ is the particle diameter; $\sigma_{g}$ is the standard deviation of the distribution; $F_{m0}$ is the number concentration as diameter $d_{gN}$, which is the number median diameter.*

*(Equation 1 included here)."*

Q: *Ensure that the range of literature values are all defined using the same definition of the lognormal function. My preference would be to use the more common definition listed above, wherein all values of Sg are >1.*

A: *All of the references from Tables 4 and 5 have been reviewed in order to check the definition of the lognormal function used. The Tables have been updated with references which used the same log-normal definition as that used by the model in this contribution. This has been clarified in the text.*

*"To cope with that, Tables 4 and 5 summarize the observed DG and SG found through a comprehensive literature review, and selecting those works using the same definition of the log-normal function described in Equation 1."*

Q: Make sure that diameter, rather than radius, is used consistently.

A: *The text has been reviewed in order to keep consistency.*

Q: Make sure all figure captions describe the contents of the figures clearly. For example, Fig. 1 caption currently says, "AOD at 550nm and differences for simulations of sensitivity test at 50%". It should say something like, "Map showing the difference between the base case and sensitivity tests using 50% changes in parameters. a) Base case showing AOD. b) Aitken mode with 50% reduction in Sg. c) Accumulation mode with 50% reduction in Sg." etc. Or have much larger, clearly defined labels on the columns and rows.

A: *The caption of the figure has been improved following the reviewer's advice.*

Q: Perform a very thorough proofreading, making sure all sentences are logical and complete and looking for grammatical errors. Perhaps Dr. Fast could help with this.

A: *English grammar was carefully revised.*